# Differential Responses of Blood Essential Amino Acid Levels Following Ingestion of High-Quality Plant-Based Protein Blends Compared to Whey Protein—A Double-Blind Randomized, Cross-Over, Clinical Trial

**DOI:** 10.3390/nu11122987

**Published:** 2019-12-06

**Authors:** Jessica L. Brennan, Maneephan Keerati-u-rai, Huaixia Yin, Julie Daoust, Emilie Nonnotte, Laurent Quinquis, Thierry St-Denis, Douglas R. Bolster

**Affiliations:** 1Danone North America, Louisville, CO 80027, USA; 2Sequel Naturals, Burnaby, BC V5G 4W3, Canada; 3Excelya, 92100 Boulogne-Billancourt, France; 4Danone Research, 91120 Palaiseau, France; 5Danone North America, White Plains, NY 10605, USA

**Keywords:** protein, plant-based protein, whey protein, essential amino acids, leucine, healthy men

## Abstract

This study assessed the bio-equivalence of high-quality, plant-based protein blends versus Whey Protein Isolate (WPI) in healthy, resistance-trained men. The primary endpoint was incremental area under the curve (iAUC) of blood essential Amino Acids (eAAs) 4 hours after consumption of each product. Maximum concentration (C_max_) and time to maximum concentration (T_max_) of blood leucine were secondary outcomes. Subjects (*n* = 18) consumed three plant-based protein blends and WPI (control). An analysis of Variance model was used to assess for bio-equivalence of total sum of blood eAA concentrations. The total blood eAA iAUC ratios of the three blends were [90% CI]: #1: 0.66 [0.58–0.76]; #2: 0.71 [0.62–0.82]; #3: 0.60 [0.52–0.69], not completely within the pre-defined equivalence range [0.80–1.25], indicative of 30–40% lower iAUC versus WPI. Leucine C_max_ of the three blends was not equivalent to WPI, #1: 0.70 [0.67–0.73]; #2: 0.72 [0.68–0.75]; #3: 0.65 [0.62–0.68], indicative of a 28–35% lower response. Leucine T_max_ for two blends were similar to WPI (#1: 0.94 [0.73–1.18]; #2: 1.56 [1.28–1.92]; #3: 1.19 [0.95–1.48]). The plant-based protein blends were not bio-equivalent. However, blood leucine kinetic data across the blends approximately doubled from fasting concentrations, whereas blood T_max_ data across two blends were similar to WPI. This suggests evidence of rapid hyperleucinemia, which correlates with a protein’s anabolic potential.

## 1. Introduction

There is increased interest in plant-based diets among consumers who consider themselves vegan, vegetarian, or lactose-intolerant, and the role of plant-based proteins specifically among active individuals and trained athletes. Protein supplementation is a common practice amongst athletes, of which animal-based proteins, such WPI are considered the “gold standard” based on its high digestibility and favorable amino acid profile [1]. Protein quality as defined by the Joint Food and Agriculture Organization of the United Nations (FAO)/World Health Organization (WHO) Expert Consultation on Protein Quality Evaluation, is calculated using the Protein Digestibility Corrected Amino Acid Score (PDCAAS), which refers to how well dietary protein can match the demand for amino acids and can predict the level of utilization of the protein [2]. This definition has been adopted by the United States Food and Drug Administration (FDA) and elsewhere globally. PDCAAS is a function of the essential Amino Acid (eAA) profile and digestibility of the protein.

Consuming adequate levels of protein, especially following physical activity, helps to optimize rates of Muscle Protein Synthesis (MPS) compared to muscle breakdown, which ultimately supports lean muscle mass accretion [3]. Additionally, the magnitude of blood amino acid response, or hyperaminoacidemia, following ingestion of protein, is an important determinant for stimulating MPS. The eAA composition of a protein relates to its ability to stimulate MPS, where those proteins having all eAA in adequate quantities would have the optimal ability to stimulate MPS [1]. The Branch-Chained Amino Acids (BCAA’s), isoleucine, valine, and leucine, are a unique class of eAA due to the role they play in supporting MPS [4]. Indeed, there are data to suggest that leucine is the most potent eAA responsible for postprandial stimulation of MPS. Thus, it is generally considered that leucine content of a protein is an important and independent predictor of its capacity to stimulate postprandial MPS [5]. 

Plant-based protein sources typically have less leucine (~6–8%) compared to animal-based proteins (>10%) [1]. Therefore, to match the leucine content of dairy proteins, individual plant-based proteins must be consumed in higher dosages (~50–60 g). Purpura et al. found that a plant-based protein source (48 g of protein from rice protein isolate, RPI) elicited similar blood amino acid responses to WPI, when provided at high levels [6]. Moreover, Gorissen et al. concluded that a plant-based protein hydrolysate (60 g of protein derived from wheat) had similar digestion and absorption patterns to animal-based proteins [7]. Collectively, animal and human studies have demonstrated that when leucine level is matched, animal-based proteins (namely, dairy proteins) and plant-based proteins have similar MPS effects [8,9].

In the diet, the consumption of a blend of plant-based proteins (i.e., complementary proteins) is a common strategy to compensate for the fact that individual plant-based protein sources are typically deficient in one or more eAA. Thus, the formulation of a plant-based protein blend with the highest PDCAAS (i.e., PDCAAS = 1.0) and similar leucine content of WPI represents an opportunity to develop a high-quality protein option, which may be advantageous to an athlete. 

We hypothesized that a high-quality plant-based protein blend with a 1.0 PDCAAS would be bio-equivalent (defined in this study as similar blood eAA response) to WPI. Therefore, the primary objective of the study was to assess the bio-equivalence of the total blood eAA response over 4 hours to the three plant-based products (Test) versus WPI product (Control). The secondary objectives were to assess the bio-equivalence and leucine kinetics over 4 hours to the three test products versus Control product.

## 2. Materials and Methods 

An acute, randomized, double-blind, 4 × 4 William square cross-over study was conducted from September to November 2018 to assess the bio-equivalence of the blood eAA response over 4 hours after consumption of 3 distinct high-quality (PDCAAS = 1.0) plant-based protein blends versus WPI in healthy, resistance-trained adult men. This study was approved on 13 September 2018 by the Western International Review Board (Puyallup, Washington, DC, USA). 

### 2.1. Participants

Participants were healthy, adult men, 18–35 years of age, Body Mass Index (BMI) between 18.5 and 29.9 kg/m^2^. Participants were required to have self-reported resistance training experience of no less than two years, with resistance training of at least one hour/day for two days/week over the past six months. Participants were instructed to abstain from protein supplements for one day prior to each of the study visits. Subjects were excluded if they had a known history of gastrointestinal, liver, kidney, or cardiovascular (including, but not limited to, atherosclerotic disease, eating disorder, myocardial infarction, peripheral arterial disease, stroke), and pulmonary disease, mental disease, seizures, use or abuse of psychoactive medications or any medication or condition which might, in the opinion of the study medical director either (1) make participation dangerous to the subject or to others, or (2) affect the results. Subjects with recent antibiotic or anabolic steroid or corticosteroids were also excluded. They maintained their habitual diet, and physical activity throughout the study. Adverse events and serious adverse events were reported throughout the whole study. 

### 2.2. Intervention

Each participant had five on-site visits, consisting of a screening phase, during which the subject eligibility was assessed, and a total of four intervention phases. During the intervention phase, each participant was studied on four separate days with the order of study products randomly assigned via a Williams Square 4 × 4 design to one of four sequences: #1#2C#3, #2#3#1C, #3C#2#1 or C#1#3#2, in which to receive the four interventions study product. Subjects crossed over to the other study product after a washout period consisting of a minimum of four days but no more than 14 days between interventions. In the day preceding each study visit, participants consumed a standardized dinner consisting of two frozen meals selected based on participant body mass, age, and activity level (total nutritionals for both meals: Hungry-Man^®^ Fajita Chicken, per serving: Calories 960, Carbohydrates 158 g, Protein 60 g, Fat 16 g), followed by a 12 hours long overnight fast. Subjects were instructed to drink water ad libitum. A 24 hours dietary recall was collected by the investigator or delegate through interview of the subject. A photocopy of the 24 hours recall collected was provided to the subject so that the diet could be duplicated before each subsequent visit.

On the morning of each study visit of the intervention phase, participants had an indwelling catheter inserted into a forearm vein by a registered nurse and the first blood sample (fasting) was collected. After the fasting sample was collected, the participant was given a study beverage mixed with 360 mL of water and instructed to consume this over 10 minutes. Additional blood samples were collected at 15 minutes, 30 minutes, 1 hour, 2 hours, 3 hours and 4 hours (+/−5 minutes) after the consumption of the study beverage. Blood samples were collected into 4 mL K_2_EDTA tubes. After a wash-out period of four to 14 days, the experiment was repeated with the participants consuming the other formulations. 

### 2.3. Study Products 

The study products consisted of dairy (Control) and plant-based proteins (3 test products) in a sweetened flavor system. The control product was a whey protein isolate (Optimum Nutrition, Downers Grove, IL, USA). All three plant-based blends included pea protein (PurisPea, Minneapolis, MN) and pumpkin protein (Austrade Inc., Palm Beach Gardens, FL, USA). Blend #2 contained, in addition to the pea and pumpkin protein, sunflower protein (Austrade Inc., Palm Beach Gardens, FL) and coconut protein (Austrade Inc., Palm Beach Gardens, FL, USA). Blend #3 represented a hydrolysis of Blend #1, in that the pea and pumpkin proteins were hydrolyzed (<15%) utilizing a commercially available, food-grade enzyme (Novozymes North America, Franklinton, NC, USA). The content of each of the study products used are displayed in Table 1 below. The plant-based blends were formulated to meet a 1.0 PDCAAS and matched the level of leucine to WPI.

### 2.4. Measurement of Blood Amino Acids

All 9 eAA were measured in the blood (as nmol/mL) (histidine, isoleucine, leucine, lysine, methionine, phenylalanine, tryptophan, threonine, valine) for 4 h (fasting, T_15_, T_30_, T_60_, T_120_, T_180_, T_240_). The blood amino acids were analyzed on a Waters Acquity UPLC System. A 200 µL aliquot of the blood was deproteinized using 190 µL of HPLC grade acetonitrile. An amount of 10 µL of 25 µmol/mL Norleucine was added as an internal standard. The solution was thoroughly vortex-mixed and centrifuged at 10 × 1000× *g* for 15 minutes to remove the precipitated proteins. Then, 40 µL of the deproteinized blood (supernatant) was transferred into a 6 × 55 mm glass culture tube and dried under vacuum using a centrifugal evaporator. After drying, the sample was treated with a redrying solution consisting of methanol: water: triethylamine (2:2:1), vortex-mixed and dried under vacuum. Then the sample was derivatized for 15 minutes at room temperature with a derivatizing solution made up of methanol: water: triethylamine: phenylisothiocyanate (7:1:1:1). After 15 minutes, the derivatizing solution was removed under vacuum. The derivatized sample was again washed with the redrying solution, vortex-mixed and dried under vacuum. The derivatized sample was dissolved in 100 µL of sample diluent (pH 7.40) and 3 µL was injected into the column, running on a modified Pico-Tag gradient using proprietary buffers (Pico-Tag Eluent 1 & Eluent 2) from Waters. The column temperature was at 48 °C. The derivatized amino acids were detected at 254 nm. The Waters Acquity Ultra Performance Liquid Chromatography (UPLC) system employed consists of a Binary Solvent Manager, a Sample Manager, a TUV Detector and a Waters Acquity UPLC BEH C18 column (2.1 × 100 mm); the relative standard deviation for the peak area using this protocol is no more than 1.5%. Data was collected, stored, and processed using Waters Empower 3 Chromatography software. Drying was done using a Tomy CC-181 Centrifugal Concentrator with an Oerlikon TRIVAC D8B Vacuum pump.

### 2.5. Outcomes

Primary endpoint was defined as the total sum of blood eAA concentration over 4 hours as the incremental Area under the Curve (iAUC). iAUC was defined as blood eAA values above the baseline value (T_fasting_). Secondary endpoints were the Leucine iAUC over 4 hours, the observed maximum concentration (C_max_) (nmol/mL) and the time (minute) to reach C_max_ (T_max_) of Leucine over 4 hours.

### 2.6. Sample Size

There were no data available regarding the expected difference between the three Test products and Control nor data regarding the expected residual error variance associated with the primary characteristic to be studied (i.e., total sum blood eAA incremental Area Under the Curve (iAUC) over 4 hours). As a consequence, the adequacy of the trial size was assessed using a range of plausible Coefficient of Variations (CV) from 15% to 35%, by steps of 5%. Using these values, the power of the trial to show equivalence for a pair of products in a 4 × 4 Williams crossover design given a sample size of 16, a desired type I error at alpha (α) level of 0.1, and two-sided with 5000 simulations keeping CVs of 15% to 20%, resulted in a power of 86–98%. We anticipated a screening failure rate of 50% and a drop-out rate of 20%; therefore, approximately 40 subjects were planned to be screened. Thus, a total of 20 randomized subjects were calculated to reach a target of 16 completed subjects based on the assumption given above.

### 2.7. Statistical Methods

Descriptive statistics overall and by randomized sequence were generated to summarize the baseline characteristics, demography, study conduct parameters (compliance to study products, study durations, consumption of forbidden dietary products and treatments). The primary outcome parameter was analyzed on the log scale with an analysis of variance (ANOVA) model with fixed effect terms for sequence, product, period and subject within-sequence fitted as a random effect. The incremental area under the curve (iAUC) above the baseline value versus time (minutes) was determined using the trapezoidal rule for each study condition over the 4 hours period following ingestion in assessing the bio-equivalence (defined here as the response of blood eAA). Next, Least Square Means (LS-Means) by study product were extracted from the analysis and back transformed to provide Geometric Least Square Means (GLS-Means). For the difference between Test and Control products, LS-Means were extracted using the estimate statement in PROC MIXED, together with the associated 90% two-sided Confidence Interval (CI). These differences in LS-Means and CIs were back-transformed to present the ratio of Test to Control GLS-Means and associated 90% CI. For bio-equivalence to be demonstrated, the entirety of the 90% CI for the ratio of Test to Control GLS-Means must lie within the range of 0.80 to 1.25. The same approach was performed for secondary endpoints (Leucine iAUC and C_max_ over 4 hours). For the Leucine T_max_, no logarithmic transformation was applied; the LS-Means were estimated using the ANOVA model described above and the 90% CI for the ratio of Test to Control was estimated with the Fieller’s theorem [10]. The analyses were performed using SAS System package (SAS Institute Inc.), Version 9.4. 

## 3. Results

Primary and secondary endpoints were reported on 18 subjects (per protocol), as illustrated by the CONSORT flow diagram (Figure 1). No significant differences between the sequence were observed in baseline and clinical characteristics at the start of the study. The subjects’ characteristics overall, and by sequence, are displayed in Table 2. The compliance was perfect; all subjects took the four study products in the order according to the planned randomization sequences and within 10 min after fasting blood sample withdrawal. 

The total sum of blood eAA iAUC over 4 hours was lower (~30% to 40%) in plant-based products compared to WPI product. Figure 2a displays the total eAA concentration by each of the conditions over the duration of the 4 hours following ingestion. In Figure 2b, the total sum of iAUC of plasma eAA over the 4 hours periods after ingestion of each of the study products is shown; all three of the plant-based protein blends had significantly different total iAUC values compared to the WPI. 

The differences in eAA between the plant-based protein blends and WPI were confirmed with the model estimates of the three ratios and 90% CI Blend #1 (pea + pumpkin): 0.66 [0.58–0.76]; Blend #2 (pea + pumpkin + sunflower + coconut): 0.71 [0.62–0.82]; Blend #3 (pea + pumpkin hydrolysate): 0.60 [0.52–0.69] when compared to WPI. Equivalence could not be concluded between any plant-based product and WPI since, in each instance, the 90% confidence interval did not fall entirely within the range of [0.80–1.25].

Leucine levels in blood over 4 hours of plant-based products versus WPI are shown in Figure 3a and the total iAUC of leucine concentrations for the duration of the study are displayed in Figure 3b by study product. The study products were not found to be bio-equivalent with respective ratios and 90% CI Blend #1: 0.66 [0.59–0.73]; Blend #2: 0.67 [0.61–0.75]; Blend #3: 0.62 [0.56–0.69]. These values are shown in Table 3. The study product by period profiles revealed that the maximal concentration observed (C_max_) over 4 hours was higher in the WPI product group with mean (SD) values between periods in a range of 647.5 (116.2) to 761.8 (142.1) nmol/mL as compared to the plant-based products where mean C_max_ were in a range of 434.1 (28.6) to 561.8 (53.8) nmol/mL (Figure 3a; Table 3). The observed time to reach C_max_ (T_max_) was numerically similar between the Blend #1, Blend #3 and WPI with mean (SD) of 42.5 (16.5), 53.3 (28.3) and 45.0 (15.4) minutes, respectively, as compared to Blend #2 with a T_max_ mean of 70.0 (29.1) minutes.

Among the 19 subjects who received at least one dose of study products, no adverse event related to the study products intake was observed in this study.

## 4. Discussion

This study represents the first human investigation in which blood eAA responses to high-quality, plant-based protein blends (PDCAAS = 1.0), matched for leucine content, were compared to whey protein. The primary findings from this study were that three plant-based protein blends were not bio-equivalent to the WPI control, as measured over 4 hours post-consumption, by iAUC of blood eAA. 

Few studies exist comparing the metabolic fate of plant-based proteins (beyond soy) to animal-based protein and those that do exist generally have been conducted on single-source plant-based proteins, for various outcomes. Purpura et al. provided subjects with 48 g of RPI or WPI and measured the total blood amino acid response over four hours. RPI showed a non-significant 6.8% lower total amino acid concentration in the blood based on AUC in comparison to WPI, indicating a similar appearance of amino acids in the blood between plant and animal-based protein. Amino acids were only measured hourly, thus capturing the earlier peak blood concentration would have been missed. In the present study, subjects were given 33–34 g of plant-based protein or 24 g WPI, with matched leucine levels (2.6 g) [6]. Compared to previous studies, we were able to significantly reduce the gram amount of protein while still matching the leucine content of the WPI utilizing a plant-based protein blend. However, the blood eAA response was not shown to be bio-equivalent to that of WPI as evidenced by a 30–40% lower in total sum eAA iAUC over 4 hours. 

A unique aspect to our study was that our protein blends were all standardized to a 1.0 PDCAAS and 2.6 g of leucine, as the leucine threshold amount that triggers the stimulation of MPS approximates between two and three grams of leucine per meal in healthy young adults [11,12,13]. Other studies using single-source plant proteins have utilized significantly greater protein quantity to match the leucine content of animal-based proteins. Reidy et al. found that when matched for leucine content, a blend of WPI with soy protein isolate was able to stimulate muscle growth to a similar extent as WPI alone. Nevertheless, the WPI group had a higher peak leucine concentration at 40 and 60 minutes post-ingestion than the WPI with soy protein isolate group. Although the intervention was not purely plant-based, this study shows that protein blends with matched leucine content to dairy protein can positively effect MPS, even with a lower post-ingestion peak leucine concentration [14]. Gorissen et al. provided 60 g wheat protein hydrolysate to match the leucine content (4.4 g) of 35 g of WPI. Despite equal leucine, WPI resulted in significantly greater blood leucine concentrations compared to wheat protein hydrolysate. However, wheat protein hydrolysate did increase myofibrillar protein synthesis rates above basal rates [7]. In the present study, we were able to provide less absolute protein than these previous studies, while matching leucine levels. Although our study did not directly measure MPS, as a surrogate measure and secondary endpoint, we measured the blood leucine kinetic response (C_max_ and T_max_). Like previous studies, the leucine concentration in the blood from our plant-based interventions was not bio-equivalent to WPI. However, an interesting finding was that the leucine T_max_ of Blend #1 and Blend #3 were similar to WPI. Additionally, data across the plant-based protein blends showed an approximate two-fold increase in leucine concentration from fasting levels. From a physiological standpoint, the leucine data provide evidence of a rapid hyperleucinemia, which is a critical response associated with postprandial MPS [1]. Future studies are required to assess the ability of high-quality plant-based protein blends to stimulate MPS.

PDCAAS is the mathematical product of the true fecal nitrogen digestibility coefficient and the eAA amino-acid profile of the protein sources [15]. We initially calculated PDCAAS scores of the plant-based protein blends to that of WPI, a value of 1.0. Given that plant proteins are deficient in one or more of the essential amino acids when compared to animal proteins, we compensated in our formulas by adding more grams of protein to the plant-based blends to increase the leucine content to match WPI [15].

Naturally occurring dietary antinutritional factors found in plant-based proteins (such as phytates, tannins, and trypsin inhibitors) have been shown to negatively impact the digestibility and bioavailability of consumed dietary protein derived amino acids [16]. However, the functional properties of food proteins can be improved by processes, such as partial enzymatic hydrolysis [17]. Gorissen et al. found that wheat protein hydrolysate was similarly digested and absorbed as micellar casein measured by stable isotopes methodology. A more transient, yet substantial postprandial increase in blood amino acid availability was observed with the wheat protein hydrolysate, even though an equal amount of whey protein resulted in a more prominent postprandial increase in blood eAA concentrations. Therefore, intact dairy protein resulted in higher blood eAA concentrations compared to a plant-based protein hydrolysate [7]. In the current study, we too implemented a hydrolysate version of a plant-based protein blend. Similarly, the mild hydrolysis (<15%) that was achieved was not significant enough to achieve bio-equivalence to the WPI. The properties of protein hydrolysates are closely related to the degree of hydrolysis (DH). Although greater hydrolysis may have promoted improved blood eAA kinetics, it typically results in negative bitterness and flavor changes [17]. Balancing organoleptic attributes and degree of hydrolysis was determined as a limitation. Future studies may investigate a higher DH with plant proteins for impact on blood eAA kinetics. 

The postprandial kinetics of dietary amino acids may have also impacted our results, as it has been demonstrated that plant-based proteins are sequestered into tissues at different rates compared to dairy-based proteins [18]. Differing amino acid composition and lower digestibility, as compared to whey, have been shown to directly impact nitrogen metabolism [18]. Bos et al. found that when compared to milk amino acids, soy amino acids were digested more rapidly and were favorably directed toward deamination pathways and liver protein synthesis. The blood amino acid concentrations rose significantly and peaked one to two hours after ingestion of soy, whereas milk caused a less pronounced rise in blood amino acid concentrations that occurred later [18]. Furthermore, animal models have found that ingestion of wheat protein resulted in higher free amino acid concentration in the liver than the ingestion of representative casein and egg mixtures [19]. Based on these data, we can hypothesize that the significant influx of amino acids after soy consumption, results in a greater increase of deamination in the liver and thus, those amino acids are less available in the blood for a shorter time, as compared to milk protein. Therefore, differences in the rate of amino acid appearance in the blood may result from the differential uptake of plant-based protein derived amino acids, which could be a reason why we saw differences in the appearance of blood eAAs in our study when compared to WPI over four hours.

## 5. Conclusions

We conclude that three high quality (defined as PDCAAS equal to 1.0) plant-based protein blends, standardized for leucine content did not achieve bio-equivalence to WPI, as measured by total iAUC of blood eAA concentrations over 4 h following ingestions. However, promising leucine kinetic data may help inform future studies. Additionally, the plant-based protein blends were safe and able to be absorbed by the blood stream with a good efficiency, thus proving to be a viable alternative to the consumption of animal proteins. Further studies may investigate the capacity, upon supplementation, to improve both sports performances and MPS, comparing the effects of plant-based protein blends and animal proteins. 

## Figures and Tables

**Figure 1 nutrients-11-02987-f001:**
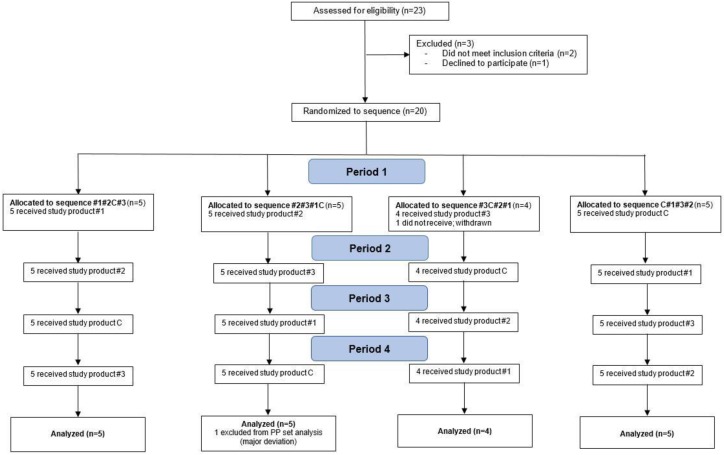
Flow of participants through the study.

**Figure 2 nutrients-11-02987-f002:**
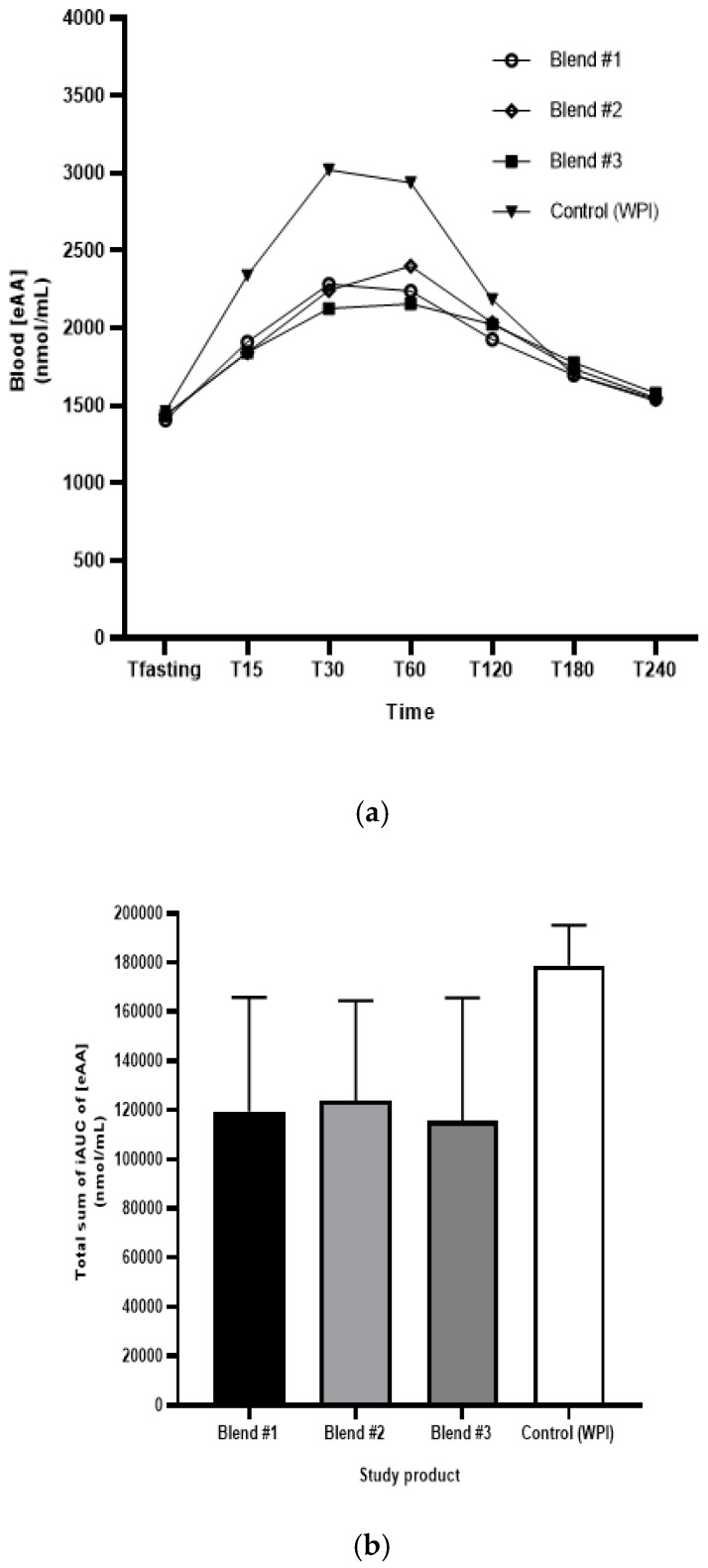
(**a**) Mean concentration of blood eAA over 4 hours following ingestion of each study product; (**b**) Mean and 95%CI total sum plasma eAA iAUC (nmol/mL) over 4 hours by study product. The area under the curve above baseline vs. time (minutes) was obtained by using the trapezoidal rule.

**Figure 3 nutrients-11-02987-f003:**
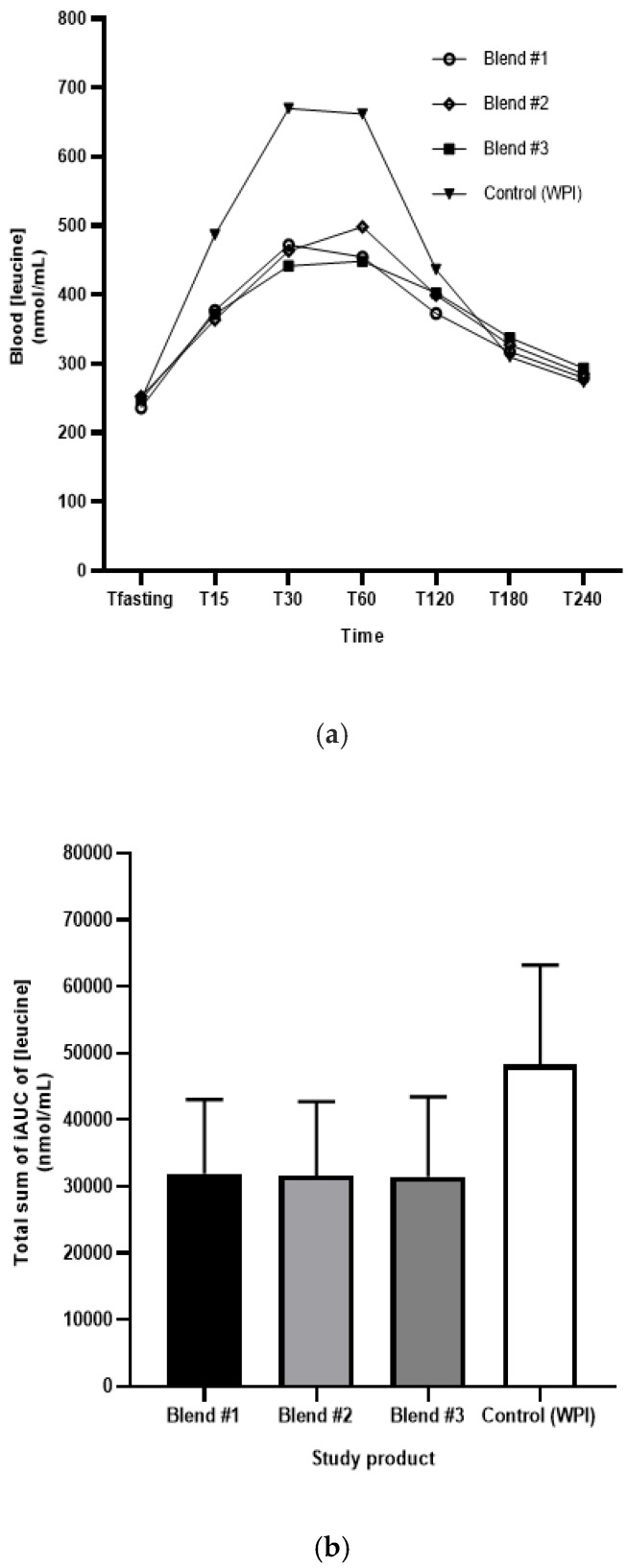
(**a**) Mean blood leucine over 4 hours per time point of each study product; (**b**) Mean and 95%CI total sum leucine iAUC (nmol/mL) by study product, obtained using the trapezoidal rule.

**Table 1 nutrients-11-02987-t001:** Composition of plant-based protein blends compared to Whey Protein Isolate (WPI).

	Study Product Comparison			
	#1	#2	#3	C
Total protein (g) for condition	34	33	34	24
Total leucine content (g)	2.6	2.6	2.6	2.6
PDCAAS	1.0	1.0	1.0	1.0
Total eAA content (g)	12	12	12	12

#1 = Protein Blend #1 (Test)—Pea, Pumpkin; #2 = Protein Blend #2 (Test)—Pea, Pumpkin, Sunflower, Coconut; #3 = Protein Blend #3 (Test)—Pea, Pumpkin (hydrolysate); C = Control—Protein Isolate (WPI).

**Table 2 nutrients-11-02987-t002:** Baseline and clinical characteristics overall and by sequence—PP population (*N* = 18).

	#1#2C#3 (*N* = 5)	#2#3#1C (*N* = 4)	#3C#2#1 (*N* = 4)	C#1#3#2 (*N* = 5)	All (*N* = 18)
Age (years)	25.2 (6.22)	27.5 (3.42)	27.5 (3.32)	22.4 (3.97)	25.4 (4.64)
BMI (kg/m^2^)	23.3 (2.79)	23.2 (5.22)	27.0 (2.26)	24.3 (2.45)	24.4 (3.35)
SBP (mmHg)	127.8 (6.38)	123.0 (11.69)	127.3 (12.28)	124.6 (9.50)	125.7 (9.25)
DBP (mmHg)	72.8 (6.06)	74.8 (9.64)	73.5 (5.80)	67.0 (10.37)	71.8 (8.13)

#1 = Protein Blend #1 (Test)—Pea, Pumpkin; #2 = Protein Blend #2 (Test)—Pea, Pumpkin, Sunflower, Coconut; #3 = Protein Blend #3 (Test)—Pea, Pumpkin (hydrolysate); BMI = Body Mass Index; C = Control—Whey Protein Isolate (WPI); DBP = Diastolic Blood Pressure; SBP = Systolic Blood Pressure. Results are displayed as mean (SD).

**Table 3 nutrients-11-02987-t003:** Leucine T_max_ and C_max_ for each study product.

Study Product	Leucine T_max_ over 4 h (min) (SD)	Leucine C_max_ over 4 h (nmol/mL) (SD)
#1	42.5 (16.5)	492.6 (47.5) *
#2	70.0 (29.1) *	508.6 (63.9) *
#3	53.3 (28.3)	462.1 (45.9) *
C	45.0 (15.4)	713.7 (105.5)

#1 = Protein Blend #1 (Test)—Pea, Pumpkin; #2 = Protein Blend #2 (Test)—Pea, Pumpkin, Sunflower, Coconut; #3 = Protein Blend #3 (Test)—Pea, Pumpkin (hydrolysate); C = Control—Whey Protein Isolate (WPI); * *p*-value < 0.001, pairwise Student *t*-test of the LS-Means Difference Tests compared to Control

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
