# Peer review of "Differential Responses of Blood Essential Amino Acid Levels Following Ingestion of High-Quality Plant-Based Protein Blends Compared to Whey Protein—A Double-Blind Randomized, Cross-Over, Clinical Trial"

_nutrients, 2019, doi:10.3390/nu11122987_

Round 1

Reviewer 1 Report

Summary: The authors characterized the blood essential amino acid response to 3 different plant-based protein supplements as compared to a whey protein isolate control matched for AA availability. The plant-based alternatives while providing eAA increasing circulating eAA were not equivalent in  concentration (Cmax or AUC) to WPI, though kinetics (Tmax) were perhaps more encouraging. However, these data provide grounds for further research.

General

As acknowledged by the authors future work is needed to determine functionality of these, or related, plant-based proteins in terms of upregulating MPS or enhancing training-induced adaptations.

The authors should provide information about the actual coefficient of variation for the blood eAA measurements. Were the samples analyzed in duplicate or triplicate?

The authors should strongly consider further study with a greater degree of hydrolysis achieved in the plant-based supplement.

Did the authors assess the gastrointestinal comfort/discomfort to the supplements? Taste? This would also provide insight into the viability of such products.

Specific

Cmax and Tmax are not defined in the abstract

Line 28-29 is the time to Cmax correlated with anabolic potential? This seems a bit overstated of the importance of kinetics vs. AUC.

Line 64 should be ‘complementary’

Line 73 why is Test capitalized?

Why did the authors choose such a large dinner?  960 kcal seems a tad on the high side.

Line 247 What is RPI? Rice protein isolate?

Author Response

As acknowledged by the authors future work is needed to determine functionality of these, or related, plant-based proteins in terms of upregulating MPS or enhancing training-induced adaptations.

The authors should provide information about the actual coefficient of variation for the blood eAA measurements. Were the samples analyzed in duplicate or triplicate?

The samples were not analyzed in duplicate or triplicate. Unfortunately, we do not have a report of the actual coefficient for the essential amino acid measurements. However, according to Hospital for Sick Children in Toronto, Canada, the lab that conducted the amino acid analyses, the relative standard distribution for the peak area is no more than 1.5%.

The authors should strongly consider further study with a greater degree of hydrolysis achieved in the plant-based supplement.

Absolutely. We agree that an interesting follow-up study would include blends with greater degree of hydrolysis. We also recognize that with greater hydrolysis, there is a potential compromise in organoleptic aspects of the protein blend including but not limited to appearance, color, taste, etc. which is certainly an aspect to consider in a future study, particularly as it relates to the palatability and acceptability.

Did the authors assess the gastrointestinal comfort/discomfort to the supplements? Taste? This would also provide insight into the viability of such products.

Unfortunately, no, we did not assess these aspects. As related to the previous comment, it is something we agree is very important concerning the viability of the blends, and certainly an aspect we considered, however, due to our financial limits, we were unable to conduct such assessment.

Cmax and Tmax are not defined in the abstract

These have been updated in the revision.

Line 28-29 is the time to Cmax correlated with anabolic potential? This seems a bit overstated of the importance of kinetics vs. AUC.

According to Phillips et al., a study conducted by Atherton et al. demonstrated that a 48 gram bolus of whey protein resulted in a 300% increase in muscle protein synthesis within 45 to 90 minutes following ingestion. Following this period, muscle protein synthesis declined despite continued availability of amino acids in circulation, which underscores the importance of the timing aspect in reaching a maximum concentration in order to maximally stimulate muscle protein synthesis, and thus, anabolism.

Phillips, B.E., Hill, D.S., Atherton, P.J. Regulation of muscle protein synthesis in humans. Curr Opin Clin Nutr Metab Care. 2012;15:58-63.

Line 64 should be ‘complementary’

This has been updated in the revised version. Thank you.

Line 73 why is Test capitalized?

This has been changed to 'test' in the revised version.

Why did the authors choose such a large dinner?  960 kcal seems a tad on the high side.

We chose an energy-dense dinner due to our participants being young, resistance-trained adult males. Thus, our primary considerations for this amount included body mass, physical activity, and age, which is now stated in the revision.

Line 247 What is RPI? Rice protein isolate?

Yes, RPI is rice protein isolate; this is defined first in line 58.

Thank you for a comprehensive review with detailed responses/comments!

Reviewer 2 Report

An interesting and well written paper. Further proof-reading to remove the grammatical errors would improve readability.

Line 155- "...the observed maximum amount (Cmax).."- Should that not be maximum concentration?

Are error bars present on figures 2a and 3a? 

Lines 316-319- "Therefore, differences in the rate of amino acid appearance in the blood may result from the differential uptake of plant-based protein derived amino acids, which could be a reason why we saw differences in the appearance of blood eAAs in our study when compared to WPI over four hours." - Two of your plant-based blends had a similar Tmax to Whey protein so did you really see such differences?

Lines 324-326- "Additionally, the plant-based protein blends were safe and able to be absorbed by the blood stream with a good efficiency, thus proving to be an invaluable alternative to the consumption of animal proteins." - Given that you have concluded that the plant blends were not bioequivalent to Whey protein, can you really conclude that they are an "invaluable alternative"?

Author Response

Line 155- "...the observed maximum amount (Cmax).."- Should that not be maximum concentration?

Yes, agreed - replaced with "concentration" 

Are error bars present on figures 2a and 3a?

We certainly can replace them with figures that contain error bars. However, our reasoning to exclude them was that, statistically, we were simply trying to illustrate the iAUC. 

Lines 316-319- "Therefore, differences in the rate of amino acid appearance in the blood may result from the differential uptake of plant-based protein derived amino acids, which could be a reason why we saw differences in the appearance of blood eAAs in our study when compared to WPI over four hours." - Two of your plant-based blends had a similar Tmax to Whey protein so did you really see such differences?

While the time (Tmax) was similar in two of the blend products compared to the whey protein, there were differences for concentrations in all three plant-based blends and Tmax in one of the blends that we felt needed to be addressed. However, little is known about the difference of uptake and handling of plant-based amino acids vs animal-based amino acids. One possibility is preferential uptake by the splanchnic bed, resulting in lower concentrations in circulation. Another possible reason for the observed differences is the presence of anti-nutritional factors within plant-based proteins which decreases the digestibility, and thus, bioavailability. In trying to account for this, we increased the amount of the plant-based blends to match the PDCAAS value and leucine content of whey protein. 

Lines 324-326- "Additionally, the plant-based protein blends were safe and able to be absorbed by the blood stream with a good efficiency, thus proving to be an invaluable alternative to the consumption of animal proteins." - Given that you have concluded that the plant blends were not bioequivalent to Whey protein, can you really conclude that they are an "invaluable alternative"?

Agreed - "invaluable" is an overstatement; changed to "viable" instead.

Thank you for your time in reviewing and providing detailed responses/comments!